# VR Human-Centric Winter Lane Detection: Performance and Driving Experience Evaluation

**DOI:** 10.3390/s25206312

**Published:** 2025-10-12

**Authors:** Tatiana Ortegon-Sarmiento, Patricia Paderewski, Sousso Kelouwani, Francisco Gutierrez-Vela, Alvaro Uribe-Quevedo

**Affiliations:** 1Mechanical Engineering Department, Université du Québec à Trois-Rivières, Trois-Rivières, QC G8Z 4M3, Canada; tatiana.ortegon@uqtr.ca (T.O.-S.); sousso.kelouwani@uqtr.ca (S.K.); 2Computer Languages and Systems Department, University of Granada, 18014 Granada, Spain; fgutierr@ugr.es; 3Faculty of Business and IT, Ontario Tech University, Oshawa, ON L1G 0C5, Canada; alvaro.quevedo@ontariotechu.ca

**Keywords:** lane detection, winter weather, virtual reality simulation, human–robot interaction, situational awareness, advanced driver assistance systems, autonomous driving

## Abstract

Driving in snowy conditions challenges both human drivers and autonomous systems. Snowfall and ice accumulation impair vehicle control and affect driver perception and performance. Road markings are often obscured, forcing drivers to rely on intuition and memory to stay in their lane, which can lead to encroachment into adjacent lanes or sidewalks. Current lane detectors assist in lane keeping, but their performance is compromised by visual disturbances such as ice reflection, snowflake movement, fog, and snow cover. Furthermore, testing these systems with users on actual snowy roads involves risks to driver safety, equipment integrity, and ethical compliance. This study presents a low-cost virtual reality simulation for evaluating winter lane detection in controlled and safe conditions from a human-in-the-loop perspective. Participants drove in a simulated snowy scenario with and without the detector while quantitative and qualitative variables were monitored. Results showed a 49.9% reduction in unintentional lane departures with the detector and significantly improved user experience, as measured by the UEQ-S (*p* = 0.023, Cohen’s d = 0.72). Participants also reported higher perceived safety, situational awareness, and confidence. These findings highlight the potential of vision-based lane detection systems adapted to winter environments and demonstrate the value of immersive simulations for user-centered testing of ADASs.

## 1. Introduction

In cities where winter snowfall scenarios are common, ice, heavy snow, sleet, and snow accumulation obstruct roadways, lanes, sidewalks, and traffic signals. This reduces mobility, increases travel time, and results in less-than-ideal visual conditions [1]. Snow affects the stability of the environment [2], causing changes in road structure and context by creating additional lane-like lines and areas within and near road boundaries [3]. Autonomous vehicles (AVs), advanced driving-assistance systems (ADASs), and human drivers face significant challenges in these extreme environmental conditions, as they struggle to accurately detect obstacles, interpret scene features, identify drivable areas, and maintain the correct lane [4,5].

Lane detection is critical for estimating vehicle position and trajectory relative to the road. It requires models capable of achieving high detection rates under diverse environmental conditions [6,7]. Both model-based [8,9,10] and deep learning-based approaches [1,11,12,13,14] have been proposed. However, lane perception algorithms degrade significantly under winter conditions. Ice, snowflakes, and fog distort sensor data, reducing accuracy and reliability [3]. For instance, LaneATT [15], an object detection-based method that uses an anchor-based attention mechanism to aggregate global information, experiences a 41.3% drop in F1 score under snow-like scenarios, where lane markings are absent or difficult to distinguish. CLRNet [16], a method that uses a cross-layer refinement network, drops 40.1% under these conditions, and UltraFast Deep Lane Detection (UFLD) [17], a method designed for challenging scenarios that represents lanes with anchor-based coordinates and learns them through classification, falls by 43.3%. This sensor and algorithm degradation increases the risk of incorrect vehicle decisions, crashes, and energy inefficiency [18].

Winter road conditions also impair human drivers’ lane-keeping performance by reducing lane boundary visibility, increasing fatigue and decreasing concentration [4,19,20]. Furthermore, confidence in AVs and ADASs may decline if technology fails under snowy conditions. Even without snow, user trust in these systems is often low; many users already show apprehension due to unfamiliarity or reported accidents. Surveys report that many potential users fear (66%) or are unsure (25%) about driving fully autonomous cars [21]. According to the American Automobile Association [22], trust improves when users receive training and guidance on how to operate AVs and ADASs.

Lane detection methods must be improved and adapted to winter conditions from both technical and human-centered perspectives. However, this requires extensive evaluation prior to implementation [23]. Real-world testing is challenging due to cost, safety risks, and ethical considerations [24,25]. Simulation provides a safe, realistic, and customizable alternative [24,26]. Technically, simulation has been used for data collection for training and validation of lane detection algorithms [27,28], systematic testing of lane-change behaviors [23], evaluation of shadow pattern-induced vulnerabilities and mitigation strategies in lane detectors [29], and analysis of perception systems under variations in sensors, lighting, and weather [30]. Human-centered studies employ simulation to assess lane-change performance [31], user reactions to takeover requests during non-driving tasks [32], unsafe driving behaviors [24], and user feedback interfaces that enhance situational awareness [33]. They have also been used to teach drivers how to interact with AVs and react to unpredictable and dangerous situations [34], and to increase trust and acceptance of AVs and ADASs through tasks such as lane changes or autonomous parking [35].

Most studies on autonomous driving and human–driver interaction are conducted in daytime and mild or temperate weather conditions, limiting discovery of new design requirements and potential solutions [36]. Environmental and road factors, and their impact on drivers’ experience, performance, AV/ADAS acceptance, and intention to use, remain understudied [37,38]. Evaluating more challenging scenarios is therefore essential to achieve a comprehensive understanding of these aspects and drivers’ responses under complex conditions [24].

Despite advances in lane detection methods, their severe performance degradation under snow (~40% drop in F1) combined with limited knowledge on drivers’ acceptance, user experience, and interaction with such ADASs under adverse conditions highlights a critical research gap. Addressing this requires safe, controlled, and immersive testing environments, such as virtual reality (VR) simulators [24,26], which allow exploration of both technical and human factors without compromising safety.

Building on previous work [39,40] where we developed a vision- and deep learning-based approach to improve lane detection on snowy roads, this paper presents a human-in-the-loop validation of winter lane detection using VR. Users’ lane-keeping performance, experience, and effectiveness of lane feedback user interfaces are evaluated in a safe, controlled, immersive, and interactive snowy driving simulation environment, monitoring variables such as heart rate, cognitive load, and vehicle-wheel contact with lane lines. Results show that the winter lane detection approach improves users’ lane-keeping performance, driving experience, and confidence under extreme winter conditions.

The contributions of this paper are summarized as follows:Low-cost virtual simulator-based evaluation in snowy conditions: We present a cost-effective virtual road environment that supports the integration and validation of vision-based lane detection algorithms in mild and snowy conditions, covering the full pipeline from virtual sensor data capture to lane visualization in a simulated vehicle. The platform also enables safe, immersive, and controlled user testing, not only allowing preliminary ADAS performance assessment but also enabling drivers to experience firsthand the benefits and practical usefulness of lane detection in challenging scenarios like snowy roads, fostering trust, acceptance, and familiarity with such systems.Influence of snow on driving performance and the role of lane detection systems: We analyze how snowy conditions affect drivers’ lane-keeping performance and user experience, and demonstrate the advantages of lane detectors adapted to these challenging conditions.User feedback on lane information presentation: The study provides insights into users’ preferences regarding lane information feedback, offering valuable data that can serve as a reference for future adaptations in real vehicles and the design of driver interfaces.

## 2. Related Work

According to the extended Dynamic Driving Task (DDT) model, the autonomous driving system performs three main activities: (i) perception, (ii) control, and (iii) actuation. This model also includes human-in-the-loop, which considers the interaction between the driver and the Object and Event Detection and Response (OEDR) components and the vehicle control [26,41].

### 2.1. Lane Detection Techniques

Accurately determining a vehicle’s position and trajectory on the road is essential, making lane detection a key component of AVs and ADASs [7]. Existing methods range from traditional model-based approaches, which geometrically model lane markings using mathematical techniques such as RANSAC or least squares regression [42,43], to deep learning approaches that leverage neural networks and can be categorized according to how they define the lane detection task as classification- [44,45], object detection- [11,12,15,46], or segmentation-based methods [14,47,48].

Although these methods perform well under standard driving conditions, their accuracy drops considerably in adverse winter environments such as snowy or icy roads. Factors such as occluded or ambiguous lane markings, reduced visibility, the scarcity of labeled datasets for extreme weather, and safety constraints during real-world snowy data collection contribute to this challenge [1,3,5,49,50,51]. To address these limitations, synthetic data generation and simulation platforms have been increasingly used to expand datasets and test lane detection algorithms under winter conditions [26,52,53]. While these approaches show promise, research explicitly targeting snow-covered roads remains limited [1,54,55], highlighting the need for further research on robust lane detection in snowy roads and other challenging scenarios.

### 2.2. Autonomous Driving Acceptance and Trust

User trust and acceptance represent critical determinants for the widespread adoption of AVs and ADASs. Several studies have proposed methods to assess these aspects, such as the Self-Driving Car Acceptance Scale (SCAS) developed by Nees [38], which evaluates user acceptance in hypothetical ownership scenarios. Nastjuk et al. [56] expanded on this by identifying the different elements, factors, or motivational patterns that shape psychological acceptance and individual adoption of AVs, based on the Technology Acceptance Model (TAM) [57]. Their findings highlight that trust, compatibility, and perceived cost positively influence usage intentions, while personal innovation, ease of use, and usefulness positively affect attitude toward the use of AVs. This attitude is a significant predictor of intention to use; however, it is negatively affected by privacy concerns. Overall, system characteristics, social influence, and personal factors play central roles, though context-specific studies remain necessary. Rahman et al. [58] further noted that direct experience with AVs tends to increase user acceptance.

Real-vehicle studies complement these insights by capturing authentic user experiences. Li et al [35] analyzed the relationship between AV-related accidents and public acceptance, proposing a multidimensional theoretical model linking vehicle reliability and the clarity of system behavior communication to psychological trust and acceptance. Ogino et al. [59] investigated automated buses and found that reducing uncertainties such as sudden stops or accelerations, and providing contextual information on the vehicle’s condition without intruding on passengers’ tasks, is crucial for social acceptance of self-driving buses. Similarly, Wang et al. [60] examined ADAS lane departure warning systems and demonstrated that tailoring warning thresholds to individual driving styles can improve system acceptance.

Trust is also closely tied to situational awareness and transparency of vehicle behavior. Lack of feedback, delayed warnings, or insufficient time for driver takeover can undermine user confidence, while effective communication fosters trust [35]. Morra et al. [61] demonstrated that interfaces enhancing transparency, such as a head-up display (HUD) providing detailed information about the AV’s state, environment, perceptions, and planned actions, can improve user confidence, situational awareness, and willingness to adopt AVs. Although such interfaces may increase cognitive load, they ultimately reduce stress and foster more positive user experiences. These findings emphasize that acceptance and trust in AVs are shaped not only by technical reliability but also by the clarity with which system behavior is communicated, a factor particularly relevant under adverse conditions such as winter driving.

### 2.3. Simulator- and VR-Based Driving Studies

On-road testing of AVs provides direct exposure to real-world conditions and captures genuine user responses, but faces limitations in safety, resources, replicability, and regulatory constraints. Simulation offers a controlled and cost-effective alternative [24,62]. Johnson et al. [63] found physiological differences such as increased heart rate in real driving compared to simulations, reflecting increased stress and highlighting both the realism and safety benefits of simulators. Simulations also support synthetic data generation for AV training and testing [52,64].

Simulation approaches span from real-vehicle setups, such as Wizard of Oz methods where a hidden driver controls the car [65] or augmented reality (AR) overlays [33], to fully virtual environments [25,26]. VR simulations, in particular, offer immersive, interactive, realistic, and cost-effective alternatives for testing human–vehicle interaction in less time, providing a strong sense of presence and safe exposure to hazardous and extreme driving situations [26]. Studies show that VR-based simulators are well accepted, practical, and effective alternatives to traditional or on-road simulations, balancing realism, immersion, and safety [25,26]. These findings highlight VR as a promising tool for both technical evaluation and human-centered research in AVs.

### 2.4. User Experience and Human–AV Interaction

VR- and simulation-based studies are increasingly used to investigate user interaction with AVs [26], assess response and performance in various driving tasks [66], evaluate user experience [25,61,67,68,69], detect unsafe behaviors [24,70], and test novel interfaces [66].

For example, Häuslschmid et al. [66] evaluated two windshield display (WSD) configurations and their impact on lane changes, measuring deviation, reaction times, user experience, and preferences between the two. Zeeb et al. [71] studied the impact of non-driving tasks on takeover performance, finding that distractions slightly affect response times while improved user understanding supports better lane keeping. These works illustrate how simulation enables systematic evaluation of human–AV interaction, providing insights into user behavior, safety risks, and interface design considerations that improve user acceptance and trust in autonomous driving technologies.

### 2.5. Influence of Environmental Conditions

Most human-in-the-loop studies in autonomous driving assume ideal conditions [36], while adverse weather such as fog, ice, and snow is rarely considered [37]. Some VR studies generated traffic scenarios under varying environmental conditions (cloudy, windy, and icy) [72,73], but focused on vehicle–environment interaction without considering the human factor.

Among the studies that explicitly address weather effects on user experience, trust, and AV acceptance, Wintersberger et al. [74] evaluated automated driving in foggy conditions using a driving simulator. They enhanced user perception by displaying occluded road objects through AR on a windshield display, using geometric shapes and color coding to indicate other vehicles’ directions, collision risks, and safe overtaking opportunities. Their results showed that providing such feedback improved trust, user acceptance, and understanding of vehicle maneuvers, but further studies are needed across diverse populations, during manual driving, and in adverse conditions.

Snow remains underexplored in terms of its impact on driver performance, user experience, and AV/ADAS acceptance. Compared to other adverse weather conditions, it poses unique challenges [75]. Unlike fog, which produces uniform and static occlusion, or rain, which mainly causes optical disturbances such as reflections or surface blurs and rarely persistently covers lane markings [76], snow can accumulate and persist, introducing complex dynamic and static effects. Dynamically, falling snow, light scattering from ice particles, irregular accumulation on the roadway, and transient features such as tire tracks or snow lines produce non-uniform visibility, where lane markings may be intermittently covered or distorted [3,5]. Statically, settled snow can conceal lane boundaries and road edges for extended periods. Additional factors such as melting and freezing cycles in sensor optics, and reduced sensing performance at low temperatures further complicate detection [77]. These conditions, and the uncertainty they introduce, not only challenge lane detection systems but can also influence human drivers’ workload, performance, overall driving experience, and trust in ADASs or AVs [4].

Despite advances in lane detection and VR-based driving studies, a critical gap remains: snow severely degrades algorithmic performance, while its impact on user performance, driving experience, and acceptance and trust in AVs and ADASs is largely unexplored. Real-world testing under such conditions is limited by safety, cost, and ethical concerns. VR-based simulations provide a safe, controlled, and immersive alternative to investigate both technical robustness and human factors, including user feedback interfaces. This study addresses this gap by evaluating winter lane detection with human-in-the-loop VR simulation. Participants’ lane-keeping performance, experience, confidence, and ADAS acceptance were measured under snowy conditions, providing insights into the technical and human-centered effectiveness of winter-adapted lane detection systems. All physiological and performance data were collected under ethics approval (CER-24-310-07.04, CER-25-324-08-01.11), anonymized, and securely stored to ensure privacy and compliance.

## 3. Materials and Methods

Lane detection in winter driving conditions is challenging for autonomous driving systems and ADASs, as well as for human drivers. The lack of visibility of road markings and the lack of situational awareness affect their perception and performance and influence the driving experience and behavior of users. From a human-in-the-loop perspective, and based on previous work [39,40], where an improved deep learning-based approach for lane detection in winter conditions was developed, in this section we present the interactive and immersive virtual snowy road simulation environment developed, along with the integration of this lane recognition approach into a virtual vehicle.

### 3.1. Driving Simulation Environment

The scenario for this study corresponds to the Trois-Rivières campus of the Université du Québec à Trois-Rivières (UQTR). It has two-carriageway roads, each with one or two lanes, and includes two roundabouts and some curved segments.

The road network was recreated using Blender software, given its versatility, affordability, resources, modeling tools, and import/export support [78]. To ensure that the 3D models were representative of the real elements that make up the scene, such as traffic signs and buildings, significant 3D authoring work was required. The layout of the road and the distribution of the different elements of the campus were obtained using a combination of OpenStreetMap and Google Maps. We built the streets using the point-to-point modeling technique and mesh editing. For sidewalks and road separators, we used the bevel curve method [79]. We modeled the campus buildings, as well as the lampposts, traffic signs, and sculptures, from primitives. To create the materials for these models, we used different photographs of the real elements as well as free textures. We modeled the lane lines from plans and placed them according to the satellite view of the UQTR campus, ensuring that they were consistent with reality. The 3D models had to be optimized to ensure an adequate number of polygons that would not cause rendering bottlenecks in VR. Figure 1 shows the reconstructed environment.

The immersive driving simulation and interactive experience with a virtual vehicle was created using the Unity 2021.3.4f1 game engine. This was chosen for its flexibility, extensibility, intuitive interface, integrated tools, game mechanics creation functionalities, resources, ease of integration with other tools such as Blender, and compatibility with immersive and interactive technologies [80]. Once the campus roads were completed in Blender, they were exported to Unity3D, where the creation of the immersive driving simulation continued. Other free-to-use 3D models were included [81,82], different weather conditions were simulated, and a virtual vehicle was added and configured, adapting it for user interaction, and integrating the previously improved lane detection system into it.

#### Environmental  Conditions

We implemented two weather conditions within our simulator: a mild environment without snow, and an environment with heavy snowfall. To recreate the snowy environment, we modified the textures of the different elements of the scene, specifically those of the streets, sidewalks, road separators, parking lots, and grass, blending them with freely available snow textures [83] using image editing software. The falling snow was simulated using a particle system in Unity. We configured the snowfall area, adjusting its shape, its size, and the number of snow particles. Regarding the flakes, we modified their scale, initial speed, and velocity over their lifetime, configuring the latter so that the snow falls randomly at different speeds. To make the flakes look more like real snow, we changed their material and assigned them random colors between white and gray, since snow is not completely white. To create the effect of snow accumulation on the road and other elements of the scene, particle collision was activated, configuring them so that they do not bounce off or slide on objects but remain on them for a certain time, thus increasing realism. To recreate the melting of the snow, the snowflakes were made to disappear with a fading effect. This effect is achieved by defining an over-lifetime color threshold, making the flakes gradually become transparent. To avoid high resource consumption, the collision of the flakes was limited to certain objects in the scene, which were assigned a layer to identify them [84]. Figure 2 shows the temperate and snowy environments.

The driving environment presented in this paper is an improved version of the version built for synthetic data collection [39], with the training and testing of the improved winter lane detection approach presented in [40]. In the present study, it is used for the safe and controlled evaluation of user experience, lane feedback user interfaces, and performance in maintaining the correct lane on a snowy road using the lane detector.

### 3.2. Virtual Vehicle

We integrated the Kia Soul EV 2017 3D model from Boisclair [85] into the virtual environment. This corresponds to a replica of the actual vehicle used at UQTR as a research platform for autonomous vehicles and energy optimization systems for intelligent transportation systems. We incorporated the steering wheel [86], the console including the instrument panel [87], seats [88], and rearview mirrors [89] into the original 3D model, and replaced the tires.

To ensure that the physical behavior of the virtual vehicle was as similar as possible to its real-life counterpart, a Box Collider was added to the car model, ensuring that it did not come into contact with the ground, as well as a Rigidbody for the physics configuration, defining a mass value for the model of 1300 kg. Wheel colliders were also added and configured to each of the vehicle’s wheels, adjusting their radius and position so that they fit around the tires. To prevent the car from overturning easily, the Unity3D default values of the suspension spring and damper were doubled [90].

To allow the user to control the vehicle and add realism to the simulation, the Logitech G29 (Lausanne, Switzerland) steering wheel, pedals, and shifter were integrated into the setup, adding the corresponding controls to ensure proper operation. The vehicle’s engine force (1000) and braking force (3000) values were set in accordance with Prism [90]. To ensure an adequate steering range, the maximum steering angle of the front tires was set at 30 degrees on each side [91]. To reflect the actual turning angle of the real steering wheel in the simulation, a maximum steering wheel turning angle of 450 degrees was set on each side, since, according to the specifications of the Logitech G29 manufacturer [92], with this model, 1.25 turns can be made on each side from the center. Besides this, we also set up the keyboard to control the vehicle using the arrow keys for steering, forward, and reverse, and the space bar for braking.

As for the rearview mirrors, we simulated the reflection effect in Unity using cameras positioned in front of each mirror, and rendered textures assigned as main maps to the mirror materials and as target textures to the corresponding cameras. To ensure that the reflection was correct, the mirror materials were flipped horizontally. Figure 3 shows the virtual vehicle model along with the elements added to it.

Given the variability in users’ body proportions, an adaptive ergonomic design was implemented in our project, enabling the driver’s seat to be adjusted to each individual’s height and leg length.

#### 3.2.1. Lane Detection Approach

Considering multi-frame approaches, early data fusion, commuting context, and image registration, in [40], we have presented an approach to improve the performance of deep learning-based lane detection models on winter roads by generating synthetic snowy road data from a small amount of ground truth data through image synthesis techniques, virtual reality, augmented reality, and simulation. Our approach provides models with richer and more generalizable information about the environment, allowing them to extract contextual, temporal, and geographical road features in both favorable driving conditions and snowy conditions. During the training phase, the models can learn and establish various relationships between lane markings and the surrounding environment under different visibility conditions, thus improving the inference of the position of the lane in winter driving scenarios.

We demonstrate the effectiveness of our data-driven approach by benchmarking four different networks fine-tuned from the Lane Shape Prediction with Transformers (LSTR) model of Liu et al. [12], trained on different real and synthetic datasets. LSTR is a parametric transformer-based model trained end-to-end with a Hungarian loss to predict a lane shape model from global context, road structures, and camera position. This network makes a bipartite correspondence between the ground truth values of the lanes and the predicted parameters to find the positive and negative lane values. Its architecture consists of a backbone, a reduced transformer network, and several feedforward networks (FFNs) for parameter predictions. The transformer consists of an encoder with two standard layers, each consisting of a self-attention module and a feedforward layer, and a decoder, which also consists of two standard layers. The self-attention mechanism models nonlocal interactions, thereby reinforcing the capture of lane structures and the learning of global information. This method is lightweight and performs well in terms of accuracy under normal environmental conditions and night scenes; however, when applied to snowy scenarios, its performance decreases dramatically [12]. When evaluating the networks, we observed a significant improvement in lane detection on snowy roads, with the best-performing model achieving an F1 score of 0.86, which was 84% higher than the F1 score obtained by the original LSTR network in these scenarios. Thus, we optimized these models in winter without needing significant modifications to existing network architectures or new network designs. Figure 4 shows the lanes predicted by the best-performing network in winter on two images of the real and virtual road of the UQTR campus.

#### 3.2.2. Integration of Lane Detection Approach into the Vehicle

The enhanced winter lane detection approach was integrated into the driving simulation environment, adapting it to recognize the lane in real time within the scene. We installed a virtual camera sensor on the hood of the vehicle, positioning it so that it pointed toward the road. For image acquisition, camera textures were used and the image resolution and bit depth were defined. Additionally, the camera was configured so that the images captured within the environment are continuously sent using a TCP protocol from Unity 2021.3.4f1 to Python 3.6.13, where the network detects the lane. The network recognizes the lanes on each of the images sent and generates a series of lane parameters corresponding to the X and Y coordinates of a set of points that define each identified lane line. Through use of a UDP protocol for higher speed, these lane coordinates are sent from Python to the Unity simulator. In a user interface located inside the virtual vehicle, the lane coordinates received from Python are plotted, overlaying them on the images of the road captured by the car’s hood camera, which are also displayed within the interface, as shown in Figure 5. The simulation was run on a computer with 32 GB of RAM, an 11th Gen Intel(R) Core(TM) i9-11900k 3.50 GHz processor (Intel Corporation, Santa Clara, CA, USA), and an NVIDIA GeForce RTX 3080 graphics card (NVIDIA Corporation, Santa Clara, CA, USA).

Due to latency constraints, executing the lane detection algorithm in real time within the VR environment was not feasible. We noticed that driving slowed down, which could disrupt visual feedback and compromise user immersion, potentially causing motion sickness or inconsistent experimental conditions. To maintain a smooth and immersive VR experience, we simulated the detector output while preserving realistic lane behavior. This approach allowed us to maintain stable frame rates and accurate lane visualization, controlled experimental timing, and ensured participants remained fully immersed during the trials. Colliders and triggers were applied to the car’s tires and lane markings to simulate detection. Several zones along the road circuit were set up, each equipped with triggers and colliders. When the vehicle’s tires enter a zone, the lanes corresponding to that road segment are displayed, as shown in Figure 6, thus simulating the progressive appearance of lanes as the vehicle moves along the road. Although this approach reduces ecological validity, as the simulated detector may not capture transient errors or delays present in a real system, it represents a balanced methodological choice for human-in-the-loop evaluations, ensuring that participants’ performance, experience, and trust are assessed under winter scenarios.

#### 3.2.3. User Feedback Interfaces

To present the identified lane to the user, we opted for bimodal feedback (visual and auditory). The auditory feedback was generated using colliders and triggers assigned to the vehicle’s tires and lane markings. When one of the front tires contacts a lane marking, a continuous warning beep is emitted, which ceases once the user returns to the correct lane.

For visual feedback, we implemented three interfaces: (i) a head-down display located at the center of the console (central HDD) as illustrated in Figure 7, (ii) another head-down display positioned on the instrument panel just behind the steering wheel (HDD behind the wheel) as shown in Figure 8, and (iii) a head-up display on the windshield (HUD). The goal was to identify the interface most accepted by users and explore possible improvements for each.

In the head-down displays, images of the road captured by the vehicle’s hood camera are displayed using a configuration similar to that of the rearview mirror, utilizing render textures and materials. The detected lanes are overlaid on these images.

For the HUD, a quad is positioned in front of the vehicle’s windshield, displaying road images captured by the hood camera with overlaid lanes. A semi-transparent material with a render texture representing the scene is assigned to this quad to aid precise alignment with the environment. Once aligned, the camera’s culling mask is configured to render only the identified lanes. This ensures that the user sees solely the lanes projected onto the windshield. Figure 9 shows this interface.

On the instrument panel, indicators were defined to inform the user of whether the lane detector is activated, as well as the vehicle’s driving mode (manual or autonomous).

#### 3.2.4. Simulation Controls

To facilitate control of the different actions and objects configured within the environment, we decided to create a command panel by programming the buttons on the Logitech G29 steering wheel, as well as certain keyboard keys. Figure 10 shows this configuration.

### 3.3. Immersive VR User Interface for Viewing the Environment

Since immersion and presence are essential for VR setups, our simulation environment employs the HP Reverb G2 Omnicept Edition VR headset (Palo Alto, CA, USA), which features non-invasive eye tracking, as well as cognitive load and heart rate monitoring. These capabilities facilitate monitoring and understanding of how test drivers behave and respond when driving under winter conditions, both with and without the lane detection system. The integration of the HP Omnicept VR headset required importation of the Windows Mixed Reality Toolkit (MRTK) and configuration of the camera game object to render the scenario to the headset [93,94].

A schematic of the simulation pipeline, from data capture to lane display, is provided in Figure 11 to clarify the workflow and enhance reproducibility.

## 4. Experimental Design

We conducted a preliminary user study adopting a human-centered mixed experimental approach, combining quantitative physiological and performance measures during driving with qualitative measures of trust, acceptance, and user experience regarding the lane detector system in snowy road conditions.

### 4.1. Ethical Approval

The study protocol involving human participants was reviewed and approved by the Ethics Committee of the Université du Québec à Trois-Rivières (Certificate No. CER-24-310-07.04) on 22 October 2024. Subsequent modifications to the protocol were also reviewed and approved under Certificate No. CER-25-324-08-01.11. Informed consent for participation was obtained from all subjects involved in the study. All physiological data collected were anonymized and securely stored to ensure participant privacy and confidentiality, in accordance with the committee’s requirements.

### 4.2. Participants

Fifteen volunteers participated in the experiment, with a wide range of driving experience, with an overall average of 12.13 years of driving (SD = 13.95). Of the sample, 40% had been driving for 10 years or more. Regarding the accident history, 40% of the participants reported having been involved in one or more traffic accidents. The familiarity of the participants with driving on snow-covered roads was limited, with only 20% having driven under such conditions.

Participant ages ranged from 19 to 64 years, with an average age of 32.9 years (SD = 15.7). Of the participants, 53.3% were older than 28 years and 46.7% were younger. The gender distribution was unbalanced, with 86.7% male and 13.3% female.

Regarding contact with autonomous systems, one participant reported having driven such a vehicle (7%), and two said that they had only had contact with them (13%). The remaining participants (80%) had neither driven nor had contact with autonomous vehicles. In contrast, participants’ familiarity with ADASs was much higher, with 60% of participants having used such systems and 7% having only had contact with them.

This resulted in a heterogeneous sample with varying levels of driving experience, age, education, and exposure to ADASs and autonomous systems, all with sufficient competence to carry out the experimental tasks.

### 4.3. Planning

We implemented a 2×2 factorial repeated-measures design with two within-subjects factors: lane detector (activated or deactivated) and snow (present or absent on the road). Each participant completed the four scenarios described in Table 1 by driving along a designated UQTR circuit. The circuit was 950 m long, with an approximate duration of 5 min per trial. The route map was presented to the participants prior to the experiment and was also displayed on the central console user interface as guidance during the virtual driving task.

For situations involving the lane detector, the three user interfaces implemented for lane visualization were activated sequentially along the circuit so that participants could evaluate them and give us their opinion. The activation of the interfaces throughout the test circuit is illustrated in Figure 12.

### 4.4. Procedure

The duration of the experiment was approximately 55 min per participant, of which 25 min was spent on driving within the virtual environment. The experiment was conducted in a single session for each participant.

#### Stages of the Experiment

Reception and characterization of the participants:At the beginning of the experimental session, the participants were welcomed and briefed on the objectives and procedures of the study, and asked to sign an informed consent form. All participants agreed to participate. Subsequently, participants completed two preliminary questionnaires. The first gathered sociodemographic information and assessed their prior experience with ADASs and automated driving systems. The second questionnaire evaluated their level of technological acceptance regarding lane detection systems.Presentation of the test and control devices:At this stage, participants were introduced to the test to be performed and the devices to be used for interacting with the simulator. They were then given time to familiarize themselves with the virtual environment and the vehicle controls. An overview of the experimental setup, with one of the participants during the test, is presented in Figure 13.Driving experience in the virtual environment:The participant undergoes the driving experience following the indicated trajectory within the virtual environment in the different driving situations. The order of the situations was counterbalanced using a Latin square to minimize order effects.End of the session:At the end of the driving experience, participants completed a series of questionnaires about their driving experience, their level of acceptance of the lane detection system, their confidence in it, and the feedback interfaces.

### 4.5. Metrics

We conducted quantitative and qualitative measurements to evaluate the impact of winter driving conditions and a lane detection system on driving performance, experience, technology acceptance, and user confidence during the driving situations described above.

To assess participants’ physiological responses across scenarios, we recorded heart rate and cognitive load using the VR headset in each situation. Table 2 provides a description of the monitored signals [95]. Driving performance was evaluated in Unity 3D by measuring the number of times participants unintentionally departed from their lane. All data were stored in log files for subsequent analysis.

In addition, various questionnaires were administered to participants to assess the aforementioned impact:Sociodemographic questionnaire: driving experience (years), age, sex, nationality, education level, and experience with ADASs and automated driving systems.Initial questionnaire on the acceptance of lane detection systems [56,96,97].Driving experience questionnaire (ease of use, acceptance, and user experience) [98,99,100,101].Questionnaire on confidence in lane detection [102].

## 5. Results and Discussion

In this section, we present the results of the preliminary user study. We analyze the data obtained to assess whether the use of a lane detection system and winter driving conditions have a direct impact on lane-keeping task performance, as well as on user experience and the trust and acceptance of this type of ADAS. Although the sample size is limited, the results provide insights into the effectiveness of the lane detection system in snowy winter conditions, both in terms of performance and user experience, as well as into the user feedback interfaces.

After starting the driving experience, two participants experienced symptoms of dizziness due to the VR headset and therefore decided not to continue the tests. The results presented below correspond to the remaining participants.

### 5.1. Driving Performance

We performed a one-way ANOVA to analyze participant performance based on the number of unintentional lane departures, evaluating overall differences between the four driving scenarios. A significant effect of the scenario on the performance was found F(3,48) = 3.45, *p* = 0.024. Tukey’s post hoc tests showed that significantly more lane departures were made in the snowy road scenario without the detector, compared to the snow-free road condition with this system (diff = 48.46; *p* = 0.025). The comparison between the snowy road scenarios with and without the lane detector showed a trend toward fewer lane departures when the detector was activated (*p* = 0.073), with approximately 41 fewer errors on average, as shown in Figure 14. Although this effect did not reach the conventional threshold for statistical significance (α = 0.05), the magnitude of the difference indicates a potential practical benefit of the lane detection system in snowy driving conditions in terms of lane keeping. A larger sample size would be necessary to confirm this effect with greater statistical power. No significant differences were observed among the other comparisons.

Subsequently, to specifically explore the separate effects of the detector, the presence or absence of snow on the road, and their interaction on the number of unintentional lane departures, we performed a two-way ANOVA with a two-by-two factorial design. A significant main effect of detector use was found (F(1,48) = 8.24, *p* = 0.006), with fewer lane departures when the detector was turned on. The sole presence or absence of snow on the road did not have a significant effect (F(1,48) = 1.67, *p* = 0.2). The interaction between snow and the detector was also not significant (F(1,48) = 0.44, *p* = 0.51), highlighting that the positive effect of the detector does not depend on the presence of snow. In both environmental conditions, lane departures are consistently reduced with the activation of this ADAS, significantly improving the performance of participants. The snow x detector interaction graph for erroneous lane departures in Figure 15 illustrates the observed trends. Although snow and interaction were not statistically significant, visual inspection suggests that the presence of snow on the road tends to increase the number of wrong-lane departures, especially when the detector is deactivated. The lane detector dampens the negative effect of snow [103,104,105].

### 5.2. Physiological Response

To evaluate the participants’ physiological response to different driving situations, we recorded their cognitive load and heart rate during these situations using the VR headset, and compared the values obtained with a one-way ANOVA and a 2×2 repeated-measures ANOVA.

Missing data were particularly present in physiological measurements due to the technical limitations of the VR headset, whose sensor accuracy was affected by unstable contact, sweat, or cosmetic interference. Given the small size of our sample (N = 15), listwise deletion would have further reduced statistical power and potentially introduced bias by disproportionately excluding participants with missing values. Mean substitution or single imputation approaches were also deemed inappropriate, as they tend to distort variance and underestimate uncertainty. Multiple imputation, despite the small dataset, provides a more principled way to handle missingness, as it preserves variability, accounts for the uncertainty of missing values, and avoids the loss of cases.

Therefore, we opted to perform multiple imputation using the mice package in R [106]. Thus, we generated five imputations using the predictive mean matching (pmm) method, which we combined to obtain a complete dataset and perform the corresponding analyses. The pmm method was used because it is particularly robust with small samples, as it imputes observed values from similar cases rather than generating purely model-based estimates. This approach is also well-suited when the variables may deviate from normality, as it does not assume a specific distribution and better preserves the variability of the original data [106,107].

#### 5.2.1. Cognitive Load

Univariate analysis of variance showed that the effect of the four scenarios on cognitive load was not significant (F(3,48) = 1.15, *p* = 0.34). Figure 16 shows the means of cognitive load per scenario, noting that these do not differ in a statistically detectable way.

The 2×2 ANOVA revealed that there were no main effects of the detector (F(1,48) = 1.25, *p* = 0.27) or presence or absence of snow (F(1,48) = 0.12, *p* = 0.73) on cognitive load. A trend toward snow × detector interaction was observed (F(1,48) = 2.08, *p* = 0.16), although this also did not reach statistical significance. Figure 17 shows the snow × detector interaction for cognitive load.

On average, participants showed moderate cognitive load in all driving conditions (0.57), with a slightly higher value in the snow-free road scenario with the lane detector (CL = 0.59) and slightly lower in the snow-free road without the lane detector (CL = 0.55). These values fall within the middle range of the cognitive load scale reported in the study by Wei et al. [108], which allows us to interpret them as close to the so-called Goldilocks zone. In this range, the mental resources invested are sufficient for participants to be attentive and engaged in the tasks, but without overloading them to the point of degrading their performance.

Although participants appear to have remained in a state of optimal effort, the results on lane maintenance suggest that contextual factors, such as the presence of snow or the use of the lane detector, affect their behavior and determine where they focus their attention. Being within the Goldilocks zone does not in itself guarantee optimal performance in this case, as the environmental features and available technological aids also have an influence.

#### 5.2.2. Heart Rate

Like cognitive load, the univariate ANOVA for heart rate also showed no statistically significant differences between the four situations (F(3,48) = 0.15, *p* = 0.93), with the measurements being almost equal considering the variability of the data, as illustrated in Figure 18.

The 2×2 repeated-measures ANOVA also revealed that the detector (F(1,48) = 0.29, *p* = 0.59) or the absence or presence of snow (F(1,48) = 0.08, *p* = 0.78) did not have significant main effects on heart rate. The interaction between the two factors was also not significant (F(1,48) = 0.07, *p* = 0.79). Figure 19 shows the snow × detector interaction graph for heart rate, suggesting that the experimental manipulation had no impact on this variable.

#### 5.2.3. Physiological Measures Discussion

Despite our expectations, heart rate and cognitive load did not show statistically significant differences across driving scenarios, including conditions with and without the lane detection system. Several factors may explain these non-significant results:Task demands: Although driving on snowy roads can be challenging, the controlled VR simulation lacked additional environmental complexity (e.g., dynamic traffic, pedestrians), which may have made the driving tasks insufficiently demanding to induce detectable changes in physiological measures.Moderate cognitive load: Participants appeared to remain within a moderate range of cognitive load (the so-called Goldilocks zone), where mental resources were sufficient to stay attentive and engaged without overloading, potentially limiting detectable changes in physiological measures.Technical and practical limitations associated with VR hardware: The HP Reverb G2 Omnicept Edition headset was used to record heart rate and cognitive load; however, sensor readings and stability can be influenced by factors such as facial sweat, cosmetic products, or improper sensor contact. During testing, the photoplethysmogram sensor occasionally failed to register data, and some participants sweated due to long (30 min) VR sessions. These factors could have reduced the sensitivity of the physiological metrics.Adaptive strategies: From a human factors perspective, participants may have employed adaptive strategies to maintain stable performance, effectively compensating for environmental challenges, which could have stabilized physiological responses even under more complex scenarios.Sample size: The small number of participants may have limited statistical power to detect subtle effects.VR context: The VR context itself may influence physiological responses differently from real-world driving. Although driving on snowy roads is demanding in real life, our simulation provided a simplified and controlled setting, lacking dynamic traffic, pedestrians, and other real-world stressors. This reduction in complexity may have lowered cognitive load and physiological stress compared to actual driving conditions.

Overall, these results suggest that while participants’ performance showed measurable differences, heart rate and cognitive load may either reflect a genuine lack of strong effects under experimental conditions or the limited sensitivity of the measurements in prolonged VR driving simulations. Caution is warranted when interpreting these metrics in similar studies. Future research could improve sensor reliability and consider alternative or additional physiological measures to more accurately assess cognitive load and heart rate in VR driving experiments.

### 5.3. Lane Detection System

#### Acceptance of the Lane Detection System

To evaluate users’ adoption of the lane recognition system and identify the factors that most influence their decision to use the ADAS, we adapted the Technology Acceptance Model questionnaire to our experiment and administered it to participants [56,57,96,97]. This questionnaire uses a 5-point Likert scale, ranging from extremely likely to extremely unlikely, and classifies the questions into three constructs: perceived usefulness (PU), behavioral intention to use (BI), and perceived ease of use (PEOU). Table 3 presents the questionnaire applied.

The questionnaire was administered twice: at the beginning of the test to understand participants’ initial concept of lane detectors, and at the end of the experiment to assess whether participants’ adoption of lane detectors changed when using our simulator.

We calculated the item correlations by construct using Cronbach’s α, Table 4, to assess the validity of the collected data.

The Cronbach’s α results from the pre-test confirm that the data are valid for re-evaluation. When comparing the pre-test with the post-test values, PU improved while BI remained at a similar value. In the case of PEOU, the change was drastic, with a decrease in α. This result does not reflect a lack of validity or poor item design, since α was high in the pre-test, but rather indicates a problem of response variability (ceiling effect) and redundancy between items.

In the post-test, participants perceived the system as easy to use, assigning scores of mainly 4 or 5 to all items, with means above 4.1 and low standard deviations (average SD = 0.29), as shown in Figure 20b. In particular, very low variability was observed in the first and last PEOU items. This homogeneity reduces the variance and, therefore, the correlations between items. In contrast, the pre-test showed a greater dispersion in responses (Figure 20a).

In analyzing the medians of the constructs together, the results after using the lane recognition system were very similar to those obtained before using it (Table 5). This suggests that the driving experience with the lane detector in the virtual environment did not produce significant changes in participants’ intention to use the detector, their perception of its usefulness, or its perceived ease of use. In the statistical analysis using Wilcoxon tests, PU showed practically identical values in the pre-test (Mdn = 4.67) and post-test (Mdn = 4.67) (V = 30.5, *p* = 0.3734, r = −0.196). BI decreased slightly after interaction with the system (Mdn = 4.2) compared to the initial measurement (Mdn = 4.4) (V = 25, *p* = 0.8121, r = 0.0102). Finally, after using the detector, PEOU was higher (Mdn = 4.5) than the initial measurement (Mdn = 4.25) (V = 40, *p*-value = 0.2146, r = −0.286); however, this difference did not reach statistical significance either.

When evaluating only the post-test acceptance results, participants reported on average a positive perception of usefulness, ease of use, and intention to use. In general, PU and PEOU showed high values with low variability, while BI was more dispersed and slightly lower. Correlation analyses between constructs revealed a moderate positive association between PEOU and BI (r = 0.42), while PU was not significantly related to BI (r = −0.07). These findings suggest that, in this context, perceived ease of use may have had a stronger influence than perceived usefulness on participants’ intention to use the system.

### 5.4. User Experience

To assess participants’ overall impression of their experience when interacting with the virtual vehicle and using the lane detector, we administered the short version of the user experience questionnaire (UEQ) [109]. The questionnaire consists of pairs of opposite attributes that the system may have, with gradations between the opposites represented by seven circles. It measures classical aspects of usability through four scales—attractiveness, efficiency, perspicuity, and dependability—of which the latter three are pragmatic quality aspects (goal-directed). It also measures user experience through two scales—novelty and stimulation—which represent hedonic quality aspects (not goal-directed). The short version of the questionnaire includes the pragmatic quality and hedonic quality scales, as well as a global scale [99]. Items are scored from −3 to +3, where values below −0.8 represent a negative evaluation, between −0.8 and 0.8 a neutral evaluation, and above 0.8 a positive evaluation [100].

By calculating the mean scores of the pragmatic and hedonic quality aspects, we observed that the user experience with the lane detector was rated extremely positively, with scores greater than 0.8 for each item. Pragmatic quality received the highest score (2.1), although hedonic quality was not far behind (1.9). Overall, the user experience was rated as excellent according to UEQ standards, with the highest scores on the perspicuity, novelty, and efficiency scales, followed by stimulation and dependability. The system and driving experience were particularly highlighted by participants as clear, interesting, easy, leading-edge, and effective. Although items such as supportive and exciting did not receive the highest scores, participants still assigned them high values, as shown in Table 6.

To obtain a clearer picture of the relative quality of the lane detector, we compared the user experience with this system to the driving experience without it. A paired-samples *t*-test was conducted with a significance level of α = 0.05, revealing that, overall, the mean scores differed statistically significantly (overall *p*-value: 0.023) and with practical relevance, showing a moderate-to-large effect size (Cohen’s d: 0.72). Both pragmatic quality (*p*-value: 0.047, Cohen’s d: 0.62) and hedonic quality (*p*-value: 0.018, Cohen’s d: 0.76) also differed significantly between the two experiences. Figure 21 illustrates this difference.

Regarding the user experience questionnaire results without the detection system, overall scores were lower, indicating a neutral evaluation, with values of 0.5 for hedonic quality and 1.07 for pragmatic quality. Perspicuity and efficiency were again the highest-scoring scales, although they received lower values than in the experience with the detector, closely followed by stimulation. The novelty scale showed the lowest performance, as expected. Participants perceived driving without the detector as somewhat obstructive and boring, although easy, as shown in Table 6.

### 5.5. Confidence in Lane Detection

To assess users’ trust in the lane detection system, we used the Situational Trust Scale for Automated Driving (STS-AD). This questionnaire evaluates different aspects of situational trust in the context of autonomous driving, taking into account six elements: trust, reaction, non-driving-related tasks (NDRTs), performance, risk, and judgment. The latter three items are scored inversely. For our experiment, we removed the NDRT element and defined the situations as driving on a snowy road and driving on a clear road. A 5-point Likert scale was used for the evaluation, ranging from strongly disagree (1) to strongly agree (5).

To assess the reliability of the adapted STS-AD version, we calculated Cronbach’s α as an indicator of internal consistency, obtaining a value of α = 0.81. This result reflects a good level of reliability, indicating that the items are adequately correlated and can be considered representative of a single construct of situational trust.

We calculated the average response to each item to assess situational trust. Overall, participants reported high situational trust in the lane detection system (3.9/5), perceiving it as reliable, safe, and competent. Positive ratings were observed for system judgment, appropriate reactions, and participant trust in both snowy and clear conditions, as shown in Table 7. This table indicates which value is considered favorable and unfavorable for each item. Performance, risk, and judgment items were reverse-scored (1 = favorable, 5 = unfavorable). Despite the inherent risk of snowy driving, the use of the ADAS led participants to perceive the scenario as neutral in terms of risk, while the non-snowy scenario was rated as non-risky. The system was rated as outperforming the user in snowy conditions, whereas performance was considered similar on the clear road.

### 5.6. Feedback Evaluation

We asked participants about the lane information feedback provided by the detection system. Overall, they agreed that it was adequate, with an average rating of 4.15 (SD = 1.14) on a 5-point scale ranging from 1 (strongly disagree) to 5 (strongly agree).

Regarding feedback channels, 50% of participants reported preferring a combination of visual and auditory feedback. A further 41.67% preferred visual feedback only, and 8.33% opted solely for auditory feedback, as they found visual feedback distracting. Concerning visual feedback, some participants suggested improvements through additional user support elements, such as a color code (green and red) to indicate lane deviation, directional markers to signal whether to move right or left to stay in the lane, or racing lines indicating the recommended speed for taking a curve without leaving the lane, which change color according to braking distance. Others suggested incorporating haptic feedback, so that the steering wheel vibrates slightly when the driver deviates from the lane.

Regarding the user interfaces evaluated, the HUD was the preferred option (69.2%), followed by the HDD located behind the steering wheel (23.1%) and finally the central HDD (7.7%). The HUD was rated as the most intuitive, easy to use, and dynamic, as it allowed participants to view the lane markings at all times without taking their eyes off the road. One participant additionally pointed out that its design appeared futuristic and suggested that the driver’s position should be considered when adjusting lane overlays to ensure accurate feedback.

In contrast, the central HDD was the least preferred by participants, as it forced them to take their eyes and attention off the road, causing insecurity and distraction, with potential risk in critical situations. The HDD located behind the steering wheel was perceived as less risky and distracting than the central HDD, although it still required glancing away from the road. Some participants also noted that the steering wheel partially obstructed the screen, making it difficult to view. Despite this, three users considered it appropriate and useful, highlighting that they found it an interesting design option.

## 6. Limitations

Although the results of our study provide valuable insights, there are some limitations that should be acknowledged.

Simulation constraints: While our VR environment provides a controlled and immersive platform for evaluating driver performance, user experience, and acceptance and trust in lane detection systems under snowy conditions, it does not yet include irregular snow accumulation, dynamic traffic, or multiple pedestrians. These simplifications were necessary in this first study to isolate the effects of snow and the winter lane detection system on lane-keeping driver performance, user experience, and system acceptance, and to avoid introducing additional variables that could distract participants and confound the results.VR-specific limitations: Simulating the lane detection algorithm instead of running it in real time introduces a trade-off in ecological validity, as participants did not interact with a real-time detector. Although the algorithm can technically run in real time, doing so slowed down the VR simulation, disrupting visual feedback and smoothness, and potentially inducing motion sickness or reducing user immersion. This decision preserved immersion and usability, while allowing for stable frame rates, accurate lane visualization, and controlled experimental timing, but may limit the direct transferability of results to real-world AV operation.Another VR-specific limitation is cybersickness, which could potentially confound stress measures. After the experiment, 53% of participants reported no dizziness, while 33% reported some discomfort. In addition, 13% of participants withdrew early due to motion sickness. Responses to the Virtual Reality Sickness Questionnaire (VRSQ) [110] indicated minimal issues for most symptoms (general discomfort, fatigue, eye strain, difficulty focusing, fullness of head, blurred vision, and vertigo), with occasional moderate or severe reports for headache and dizziness with eyes closed. These results suggest that, while cybersickness was generally low, it should be considered when interpreting psychophysiological or performance outcomes in VR studies.Sample limitations: The sample size of this preliminary study is small (N = 15), which limits statistical power and the generalizability of the findings. Although participants varied in age (19–64 years), driving experience (ranging from a few years to over 45), and educational background (from high school graduates to doctoral students), the gender distribution was unbalanced, with a predominance of male participants. The limited sample size may also have affected the sensitivity of certain measures. While differences were evident in performance, user experience, and trust/acceptance of the ADAS, physiological measures such as heart rate and cognitive load showed minimal variation across conditions. This could reflect either a genuine absence of strong effects or insufficient sensitivity due to the small sample. We acknowledge that the findings should be interpreted as preliminary and exploratory, highlighting trends and potential practical effects of the snowy lane detection system that warrant confirmation in larger and more diverse samples.Ethical protocols: The study protocol was reviewed and approved by the Ethics Committee of the Université du Québec à Trois-Rivières (CER-24-310-07.04, CER-25-324-08-01.11), and participants provided informed consent. All participant data, including physiological records, were anonymized and stored securely, ensuring privacy.

## 7. Conclusions and Future Work

In this study, we presented the design of a virtual driving environment for the safe and controlled evaluation of vision-based lane detection algorithms in winter driving conditions, from a human-in-the-loop perspective. We integrate and validate a vision- and deep learning-based approach previously implemented to improve winter lane detection, and we monitored both quantitative and qualitative variables to examine the influence of contextual factors, such as snow presence or activation of the lane detection system, on user experience, lane-keeping performance, and user trust and acceptance of this type of ADAS.

The results suggest that activating the lane detector has a significant effect on user performance, reducing unintended lane departures. This finding highlights the system’s relevance, particularly under adverse weather conditions such as snow, where it compensates for performance deterioration due to low road visibility. Thus, the lane detector is supported as a resource that enhances both vehicle control and road safety.

Although physiological measurements did not show significant differences between scenarios, they provided valuable information on participants’ concentration, attention, readiness, stress, and relaxation levels. This underlines the importance of including both direct performance measures and psychophysiological indicators to gain a more comprehensive understanding of the effects of technological aids, task demands, and the environment. In addition, the ability to interact directly with this ADAS within the immersive environment contributed to increased user trust in the system and improved their driving experience on snowy roads, evidencing a highly positive user experience given the clarity, simplicity, effectiveness, and leading-edge nature of the detector under such conditions.

### Future Work

We acknowledge that real-world driving involves additional variables not fully reproduced in our VR simulation. Future work will extend the VR environment to include more realistic conditions, such as dynamic traffic, irregular snow accumulation, multiple pedestrians, and additional weather variations. The lane detection algorithm, already integrated into the simulation, will be fully activated in real time using optimized or hardware-accelerated implementations. These enhancements aim to increase ecological validity, preserve user immersion, and allow a more comprehensive assessment of human–ADAS interaction under complex real-world winter driving conditions. Additional directions include integrating and testing additional user support elements, such as guidance lines and dynamic visual or auditory signals, with personalization mechanisms to optimize safety and user experience.

Driver responses will be assessed using multidimensional psychophysiological indicators, including EEG, skin conductance, and eye tracking, enabling a more comprehensive evaluation of drivers’ psychological and physical states and providing a richer, more reliable understanding of their behavior. To improve robustness and generalizability, future studies will involve larger and more gender-balanced participant samples.

Finally, we aim to investigate the path toward real-world deployment of the winter lane detection system, addressing hardware requirements, latency constraints, and performance under mixed or partially snowy conditions. Building on these insights, VR-based evaluations will be complemented with pilot testing in controlled real-road winter scenarios or augmented reality-based simulations, bridging the gap between virtual simulation and real-world evaluation, and ensuring that results from the immersive environment translate effectively to practical driving conditions. Direct quantitative comparisons of our method’s performance with commercial lane-keeping assistance systems are also planned, once resources, safety protocols, and ethical approvals are in place. This will strengthen the validity and contextual relevance of this study.

## Figures and Tables

**Figure 1 sensors-25-06312-f001:**
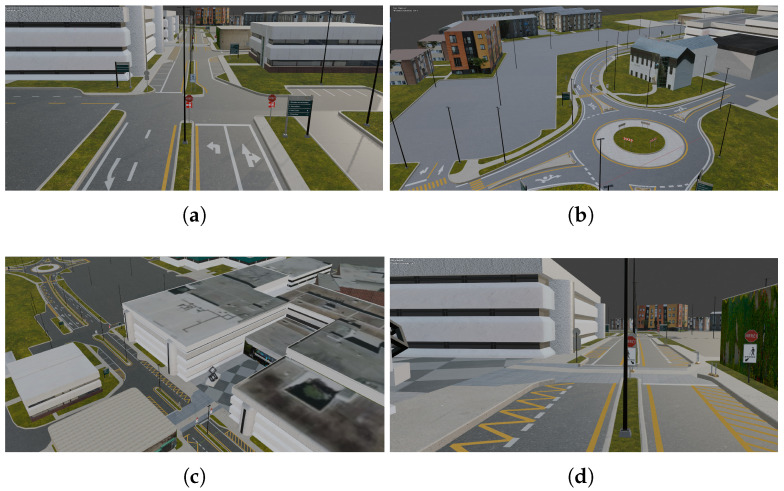
UQTR campus 3D reconstruction. (**a**) View of a road segment with two carriageways with two lanes. (**b**) View of one of the roundabouts. (**c**) Campus building complex. (**d**) View of a road segment with two carriageways with one lane.

**Figure 2 sensors-25-06312-f002:**
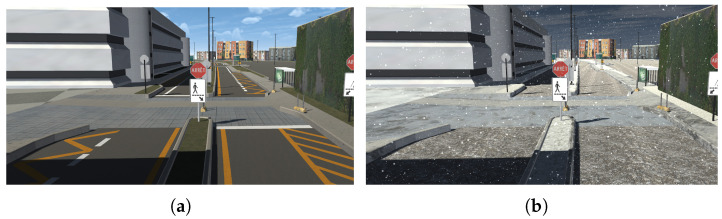
Simulated environmental conditions. (**a**) Road without snow. (**b**) Road with snow.

**Figure 3 sensors-25-06312-f003:**
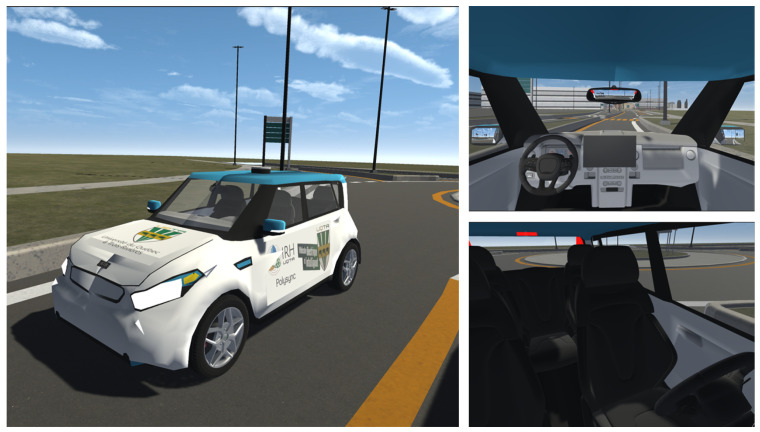
Three-dimensional model of the vehicle.

**Figure 4 sensors-25-06312-f004:**
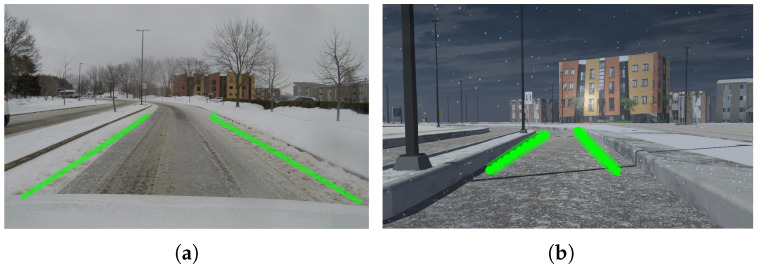
Use of the enhanced LSTR-based lane detection approach on a set of snowy road images. (**a**) Real environment. (**b**) Virtual environment. Detected lanes are shown in green.

**Figure 5 sensors-25-06312-f005:**
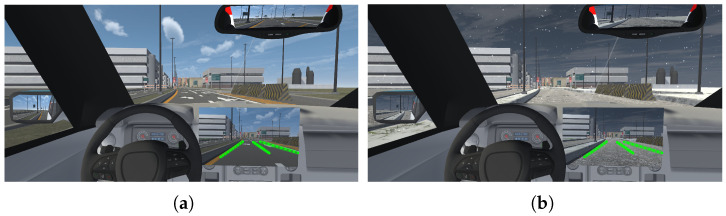
Display of identified lanes in green on the user interface configured within the virtual vehicle. (**a**) Snow-free road. (**b**) Snow-covered road.

**Figure 6 sensors-25-06312-f006:**
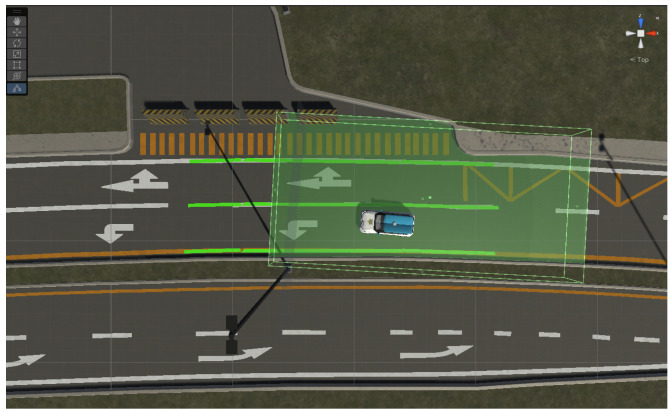
Simulated lane detection system due to the high latency generated by the use of the real detector while driving.

**Figure 7 sensors-25-06312-f007:**
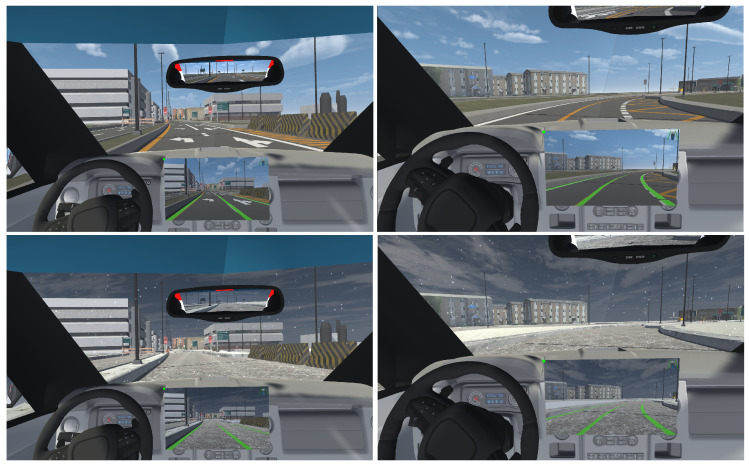
Central HDD. Detected lanes are shown in green.

**Figure 8 sensors-25-06312-f008:**
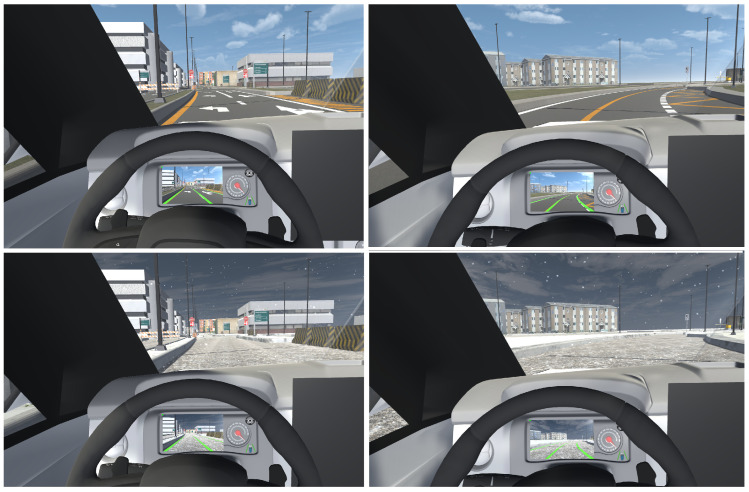
HDD behind the wheel. Detected lanes are shown in green.

**Figure 9 sensors-25-06312-f009:**
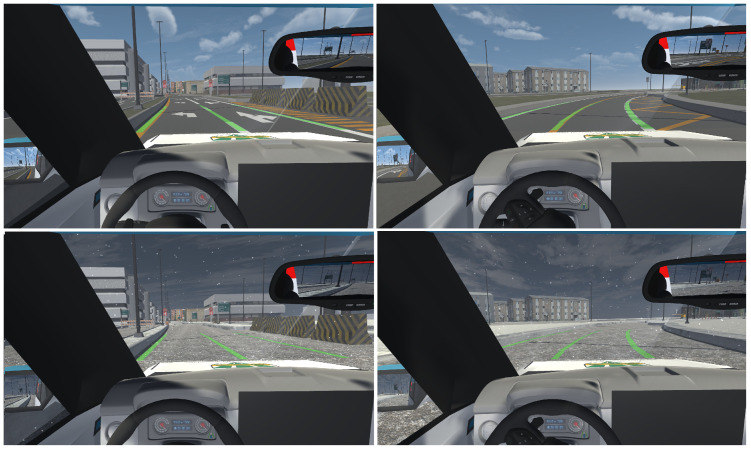
HUD. Detected lanes are shown in green.

**Figure 10 sensors-25-06312-f010:**
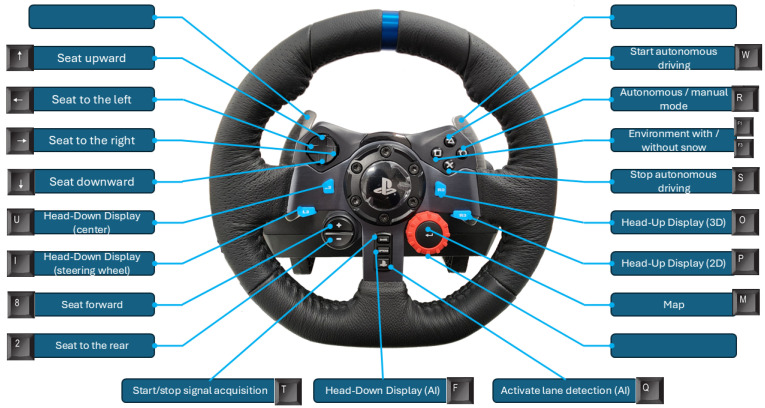
Command panel for controlling actions and objects within the driving simulator.

**Figure 11 sensors-25-06312-f011:**
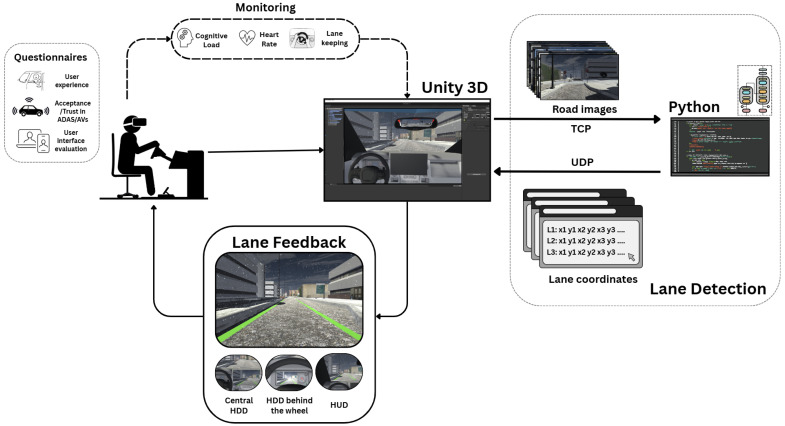
Simulation pipeline diagram.

**Figure 12 sensors-25-06312-f012:**
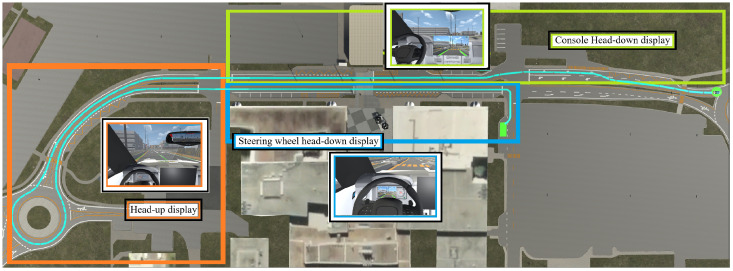
Route that participants had to follow in the virtual environment, and distribution of user interface activation. Colored boxes indicate the type of interface active in each segment: orange for HUD, blue for HDD behind the wheel, and green for central HDD.

**Figure 13 sensors-25-06312-f013:**
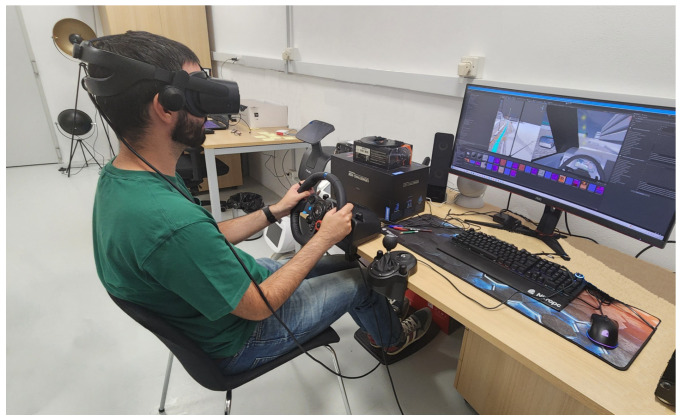
Experimental setup.

**Figure 14 sensors-25-06312-f014:**
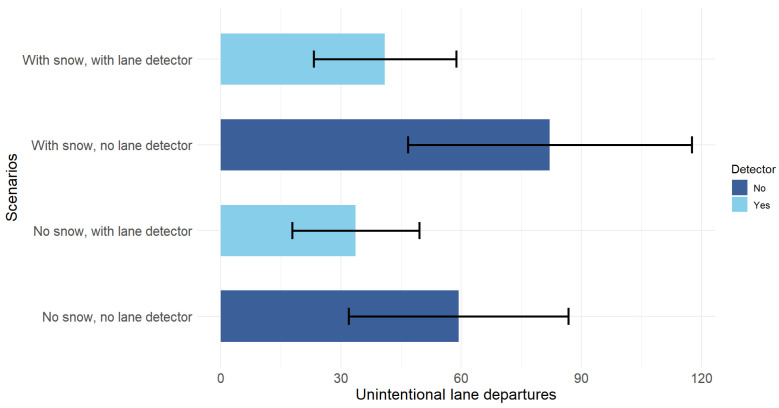
Mean unintentional lane departures for each scenario. Error bars represent 95% confidence intervals.

**Figure 15 sensors-25-06312-f015:**
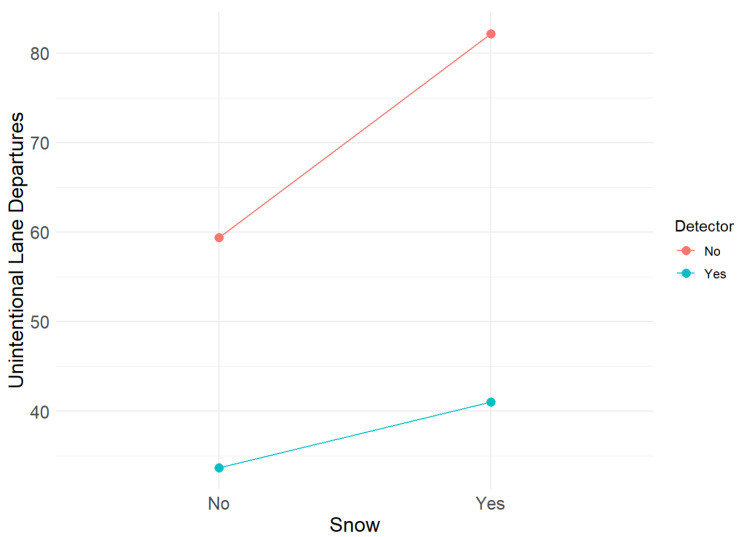
Snow × detector interaction in unintentional lane departures.

**Figure 16 sensors-25-06312-f016:**
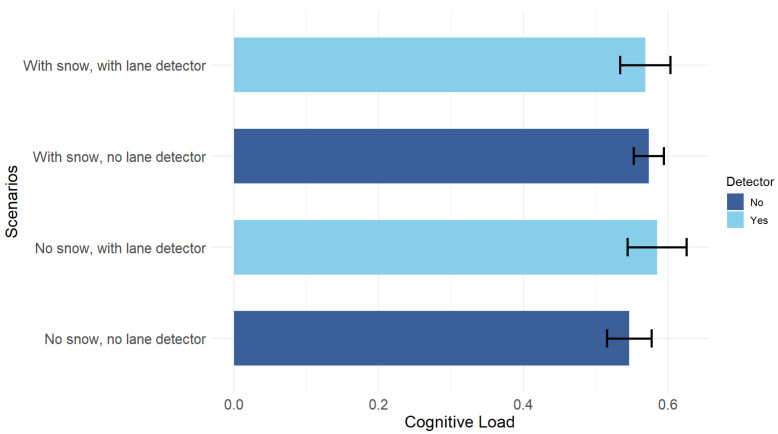
Mean cognitive load per scenario. Error bars indicate the 95% confidence intervals.

**Figure 17 sensors-25-06312-f017:**
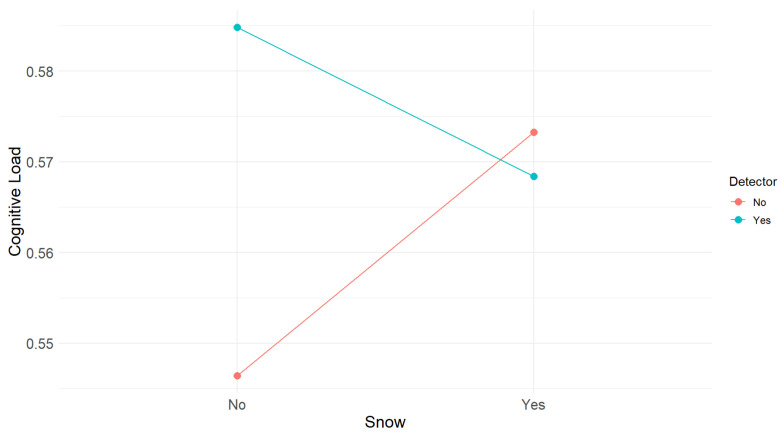
Snow × detector interaction in cognitive load.

**Figure 18 sensors-25-06312-f018:**
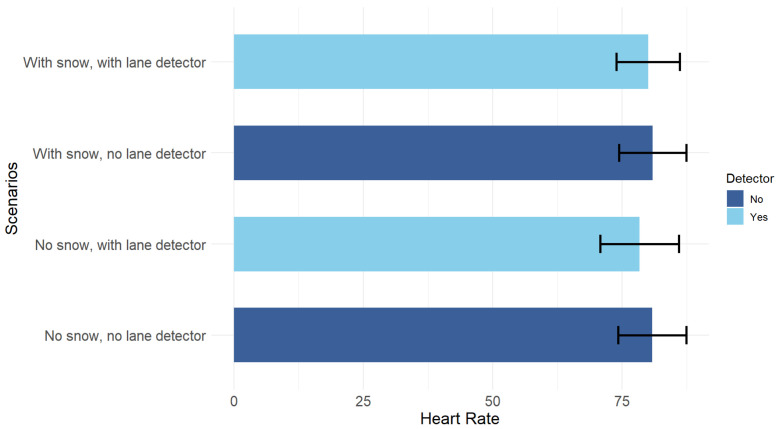
Mean heart rate for each condition. Error bars show the 95% confidence intervals.

**Figure 19 sensors-25-06312-f019:**
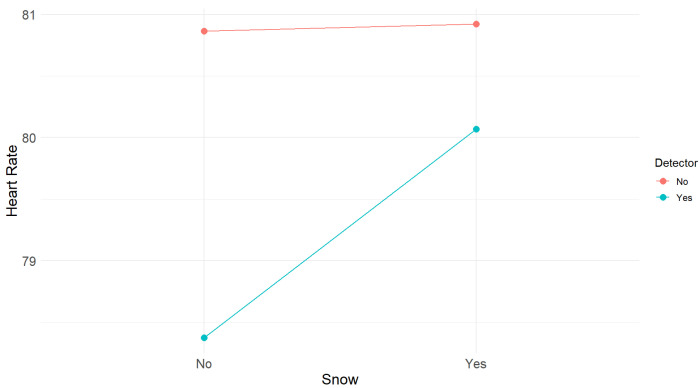
Snow × Detector interaction on heart rate.

**Figure 20 sensors-25-06312-f020:**
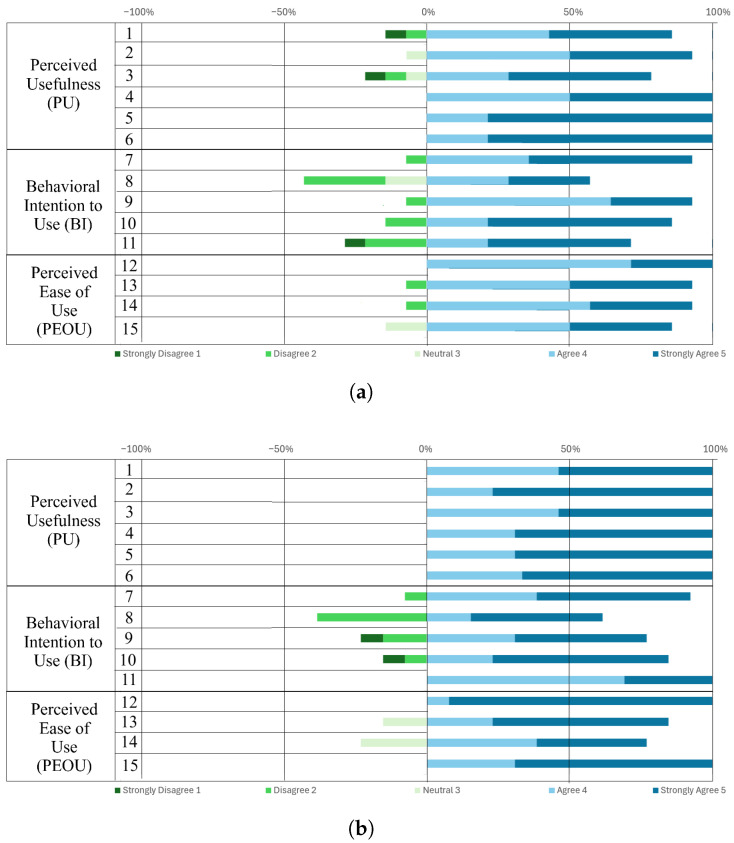
Percentage distribution of responses per item. (**a**) Pre-test. (**b**) Post-test.

**Figure 21 sensors-25-06312-f021:**
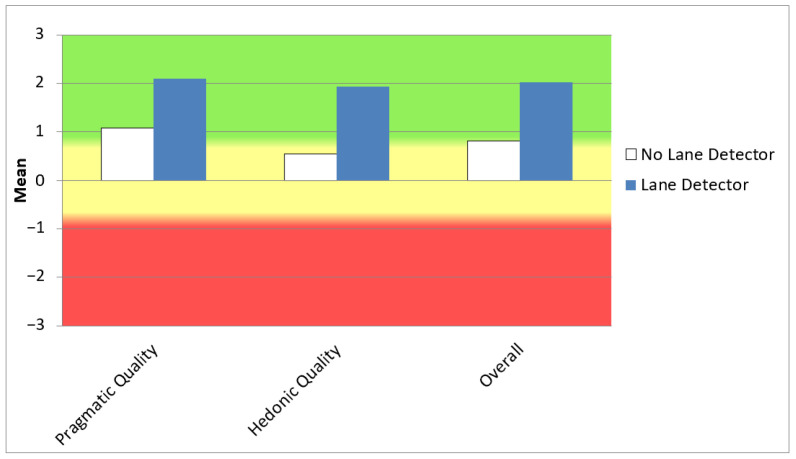
UEQ comparison of the driving experience with and without the lane detector implemented. Source: own elaboration using the UEQ data analysis tool [100].

**Table 1 sensors-25-06312-t001:** Description of experimental driving situations.

	Driving Situation	Feedback Interfaces	Estimated Duration (min)
A	No lane detection system and snow-free road		5
B	No lane detection system and snow-covered road		5
C	Lane detection system and snow-free road	X	5
D	Lane detection system and snow-covered road	X	5

Note: A–D correspond to the four driving scenarios tested. “X” indicates the presence of the feedback interface.

**Table 2 sensors-25-06312-t002:** Physiological signals monitored during the study.

	Description
Cognitive load	Cognitive load refers to the mental effort a person expends when performing a task or participating in a learning process. It can be an indicator of performance. Based on data from virtual reality headset sensors (eye tracking, heart rate, photoplethysmography (PPG)), a real-time biometric indicator of cognitive load is obtained, with a continuous value estimated between 0.0 and 1.0, where values close to 0 indicate a low load and values close to 1 indicate high load. For optimal user performance, this load should be in the midpoint of the cognitive load scale, in the “Goldilocks” zone.
Heart rate	The VR headset is equipped with a photoplethysmogram sensor located on the forehead and represented by two green LEDs, which uses light signals reflected on the skin to detect changes in blood volume in the microvascular bed of tissue. Thanks to this sensor, the headset displays the heart rate once every 5 s.

**Table 3 sensors-25-06312-t003:** Technology Acceptance Model questionnaire adapted to our experiment.

Constructs	Scale Item
Perceived Usefulness (PU)	Using the lane detection system while driving would allow me to complete tasks more quickly.
	Using the lane recognition system would improve my driving performance.
	Using the lane detection system while driving would increase my productivity.
	Using the lane recognition system would improve my driving efficiency.
	Use of lane recognition system would make driving easier.
	I would find the lane detection system useful while driving.
Behavioral Intention to Use (BI)	I will consider using the lane detection system.
	I plan to use the lane detection system.
	I would continue to use the lane detection system frequently if I could.
	I will inform others about the quality of the lane detection system.
	I felt very confident using the system.
Perceived Ease of Use (PEOU)	It would be easy for me to learn how to use the lane detection system.
	My interaction with the lane detection system is clear and understandable.
	I would say that the lane detection system is flexible and can be interacted with.
	I find the lane detection system easy to use.

**Table 4 sensors-25-06312-t004:** Correlations of items per construct.

Construct	Items	Cronbach’s α—Pre-Test	Cronbach’s α—Post-Test
PU	6	0.67	0.83
BI	5	0.78	0.7
PEOU	4	0.87	0.21

**Table 5 sensors-25-06312-t005:** Descriptive statistics of TAM for pre- and post-test.

Construct	Pre-Test			Post-Test		
	Median	Mean	Std. Dev	Median	Mean	Std. Dev
PU	4.67	4.48	0.48	4.67	4.65	0.38
BI	4.40	4.09	0.87	4.20	4.11	0.82
PEOU	4.25	4.34	0.495	4.50	4.55	0.29

**Table 6 sensors-25-06312-t006:** UEQ scores: driving with vs. without the lane detection system.

Item		Scale	Quality	Mean	
Negative	Positive			Lane Detector	No Lane Detector
Confusing	Clear	Perspicuity	Pragmatic	2.5	1.2
Not interesting	Interesting	Stimulation	Hedonic	2.4	1.2
Complicated	Easy	Perspicuity	Pragmatic	2.3	1.3
Usual	Leading-edge	Novelty	Hedonic	2.2	−0.2
Conventional	Inventive	Novelty	Hedonic	2.0	0.1
Inefficient	Efficient	Efficiency	Pragmatic	2.0	1.2
Obstructive	Supportive	Dependability	Pragmatic	1.6	0.6
Boring	Exciting	Stimulation	Hedonic	1.2	1.0

**Table 7 sensors-25-06312-t007:** Results of the adapted STS-AD with a 5-point Likert scale.

	Items	Favorable Value	Unfavorable Value	Mean
1	I trust the system in this situation	5	1	4.08
2	The lane detection system reacted appropriately to the environment	5	1	4.25
3	I would have performed better than the lane detection system on the snowy road ^1^	1	5	2.23
4	I would have performed better than the lane detection system on the road without snow ^1^	1	5	3
5	The snowy situation was risky ^1^	1	5	3.5
6	The situation without snow was risky ^1^	1	5	2.31
7	The lane detection system made an unsafe judgment in the snow ^1^	1	5	1.67
8	The lane detection system made an unsafe judgment without snow ^1^	1	5	1.42

^1^ Inverted items, where 1 represents a favorable value for confidence and 5 represents an unfavorable value.

## Data Availability

Data is unavailable due to privacy restrictions.

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
