# Peer review of "VR Human-Centric Winter Lane Detection: Performance and Driving Experience Evaluation"

_sensors, 2025, doi:10.3390/s25206312_

Round 1

Reviewer 1 Report

Comments and Suggestions for Authors

  1. The abstract would be significantly strengthened by including key quantitative results (e.g., the observed reduction in lane departures) to substantiate the claims of improved performance and user experience. It should also more clearly state the research gap their work fills.
  2. The introduction needs a sharper, concise problem statement. It should also include quantitative data on sensor degradation and cite recent review papers to better frame the problem within the existing literature. Please see the following paper as an example:
  • A systematic review of abnormal behaviour detection and analysis in driving simulators, Transportation Research Part F: Traffic Psychology and Behaviour, Volume 109, 2025, Pages 897-920, ISSN 1369-8478, https://doi.org/10.1016/j.trf.2025.01.002.
  1. The related work section should be restructured into thematic subsections (e.g., Lane Detection Tech, VR in Driving Studies, User Trust & Acceptance) to improve readability. Additionally, the paper needs to explicitly differentiate this paper's contribution from prior VR studies and address ethical considerations (data privacy) of using physiological biometrics in VR.
  2. The description of the simulation's realism and validation is vague. A visual aid (e.g., flowchart/diagram) is suggested to clarify the complex pipeline from data capture to lane display. The choice to simulate the detector (due to latency) is a major methodological compromise that needs a stronger justification and discussion of its impact on ecological validity.
  3. The paper appropriately acknowledges the limitation of a small sample size on page 16, but does not elaborate on the implications of this limitation on the generalizability.
  4. The ANOVA results are clearly presented, but the lack of significant findings in physiological measures (cognitive load, heart rate) calls into question the sensitivity of these metrics in VR contexts.
  5. The use of multiple imputation is methodologically rigorous, but the justification for its use should be expanded, given the small dataset.
  6. The discussion did not address the non-significant physiological results effectively—why might heart rate and cognitive load not have changed? It should also engage more deeply with human factors theories (situational awareness, cognitive load) to interpret the results.
  7. The limitations section should be expanded to include:
    • VR-specific limitations: Cybersickness (which already caused dropouts) and its potential to confound stress measures.
    • Sample limitations: The very small size and severe gender imbalance (87% male).
    • Ethical protocols: Explicit statement of IRB approval and how biometric data privacy was handled is mandatory for publication.

10. The conclusion should discuss the path to real-world deployment, including challenges like hardware requirements, latency, and performance in mixed or partial snow conditions.

Author Response

Dear reviewer,
The authors would like to thank you for your time to review this manuscript, your comments, and recommendations. Please see the attached file for detailed responses.

Reviewer 2 Report

Comments and Suggestions for Authors

The proposed method proposed a deep learning lane detection algorithm based on virtual reality to solve the problem that the driving stability of both the driver and the autonomous driving assistance system deteriorates due to the obstruction of lanes when driving in the snow. 

However, the proposed method has limitations in that vr simulation experiments do not sufficiently reflect complex variables on the actual road (irregular accumulation of snow, movement of other vehicles, etc.). 

In addition, the difference in heart rate or cognitive load tested to see if the driver is nervous or concentrated during driving work is not statistically significant, so it is difficult to support the argument that the lane detection system reduces the physiological and cognitive burden.

Therefore, the following improvements are required.

In addition to vr-based evaluation, it is necessary to supplement reality and immersion by performing real road driving(pilot testing) at the same time or introducing augmented reality-based simulations.

In addition to the driver's cognitive load and heart rate, it is necessary to more accurately evaluate the driver's psychological and physical reactions by integrating multidimensional indicators such as eeg, skin conduction, and eye tracking.

In addition, it is required to directly compare the quantitative performance of the proposed scheme with the existing commercial Adas lane assistance system.

Author Response

(The authors gave the same response as above.)

Round 2

Reviewer 1 Report

Comments and Suggestions for Authors

1.    Section 2.5 (Environmental Conditions) feels slightly underdeveloped compared to Sections 2.1–2.4. Please briefly expand on why snow is uniquely challenging (e.g., dynamic occlusion vs. static fog).
2.    Consistency: Use "snowy condition" vs. "snow condition" (line 851) uniformly (e.g., "snowy conditions" preferred). 

Author Response

Dear reviewer,
Thank you for your valuable feedback in this second round. Please see the attached file for detailed responses to your comments.
